# An assessment of potential improvements in social capital, risk awareness, and preparedness from digital technologies

Tommaso Piseddu[1], Mathilda Englund[1], Karina Barquet[1]

[1]Stockholm Environment Institute, Linnégatan 87D, 115 23 Stockholm, Sweden

5   *Correspondence to*: Tommaso Piseddu (tommaso.piseddu@sei.org)

**Abstract.** Contributions to social capital, risk awareness and preparedness constitute the parameters against which applications of digital technologies in the field of disaster risk management should be tested. We propose here an evaluation of four of these, mobile positioning data, social media crowdsourcing, drones and satellite imaging, with an additional focus on acceptability and feasibility. The assessment is carried out through a survey disseminated among stakeholders. The frame of
10   the analysis also grants the opportunity to investigate to what extent different methodologies to aggregate and evaluate the results, the CRITIC model, the dCRITIC model, the Entropy model, the Mean Weight model and the Standard Deviation model, may influence the preference of one technology over the others. We find that the different assumptions on which these methodologies rely deliver diverging results. We therefore recommend future research to adopt a sensitivity analysis that considers multiple and alternatives methods to evaluate survey results.

30

# 1 Introduction

The Sendai Framework for Disaster Risk Reduction (UNDRR, 2015) calls for investments in digital technologies and tools to enhance societal resilience. Recent developments in digital technologies and tools offer emerging opportunities for managing disaster risk, i.e., the potential for loss or damages determined by the function of hazard, exposure, and vulnerability (Disaster risk, 2023). More specifically, digital technologies and tools hold significant potential in strengthening social capital, risk awareness, disaster preparedness, and, in the end, societal resilience (Latvakoski et al., 2022).

Many scientific fields adopt the concept of resilience (Alexander, 2013), including ecology (Holling, 1973), psychology (Garmezy et al., 1984), and disaster research (Manyena, 2006). As a consequence, resilience is subject to diverse definitions and conceptualizations (see for example IPCC, 2014; Johansen et al., 2017; Joseph, 2018; Manyena, 2006; Morsut et al., 2021; UNDRR, 2015; Zhou et al., 2010). Some researchers suggests that resilience refers to the ability of a system to bounce back to its equilibrium (Capano and Woo, 2017; Jurgens and Helsloot, 2018). Other researchers, however, denotes the bounce back metaphor as it fails to capture changes in the social fabric that occur in the wake of a disaster (Dufty, 2012). Accordingly, resilience refers to the ability of a system to bounce forward to a new normal i.e., anticipate, recognize, adapt to and learn from societal disruptions and disasters (Becker, 2014).

There is a plethora of factors that enable or constrain resilience (Jordan and Javernick-Will, 2012). In disaster research, social capital has emerged as a critical determinant of resilience (Kerr, 2018). Social capital refers to "features of social organizations, such as networks, norms and trust that facilitate action and cooperation for mutual benefit" (Putnam, 1993, p.35).

Greater levels of social capital within a community are linked to higher levels of disaster preparedness and risk awareness (Brunie, 2007; Hausman et al., 2007; Morsut et al., 2021). The nexus between social capital, risk awareness, and disaster preparedness can improve and facilitate collaboration; provide social safety nets; strengthen communication and information-sharing; speed up response and recovery efforts; and in the end improve resilience among the most vulnerable segments of the population (Aldrich and Meyer, 2015).

Despite the demonstrated importance of risk awareness, social capital, and disaster preparedness for effective disaster risk management, their relative degree of importance for enhancing societal resilience remains unclear. An array of weighting models exist, all of which can produce different results (Mukhametzyanov and Pamučar, 2018; Odu, 2019). Most quantitative vulnerability assessments employ a mean weighting approach by default, due to the limited knowledge of the relationship between indicators (Tate, 2012). Comparative studies combining or juxtaposing different weighting methods are found in other fields of research (Jayant and Sharma, 2018) and conclude by warning against the belief that employing one single method may provide sound conclusions (Botvinik-Nezer et al., 2020; Wagenmakers et al., 2022). Similar studies are overall lacking when it comes to digital technologies and tools for improving societal resilience and disaster risk management.

In this paper, we evaluate digital technologies and tools for improving social capital, risk awareness, and disaster preparedness by using a multicriteria analysis. We thereafter assess to what extent the findings are sensitive to different weighting methods. We use five different weighting methods: CRITIC, dCRITIC, Standard Deviation (SD), Mean Weights (MW), and Entropy.

Final scores for these digital technologies are computed using two different approaches, the Weighted Sum Approach (WSA) and the Product Sum Approach (PSA) to the test to what extent the results are dependent on this last computational step. This research builds on work carried out within the Horizon2020 BuildERS project that aims to improve social capital, risk awareness, and preparedness among the most vulnerable segments of the European population through digital technologies and research.

The rest of the paper is organized as follows: Section 2 introduces the digital technologies we focus on, Section 3 describes the methodology and the data, Section 4 presents the results, Section 5 comments on the results and Section 6 concludes.

## 2 Digital technologies in disaster risk management

Here, digital technologies refer to "tools, systems and devices that can generate, create, store or process data" (Johnston et al., 2022, p.n.d.). Digital technologies can support i) sensing, i.e. collect data with remote sensors, ii) communication, i.e. transmit
data, iii) processing, i.e. modelling and analysis, and iv) actuation, i.e. data reconstruction (Bao et al., 2022). Digital technologies can be applied throughout the whole disaster risk management cycle – mitigation, preparedness, response and recovery (Sakurai and Murayama, 2019; Vermiglio et al., 2022). Digital technologies such as smartphones have become ubiquitous in modern life, but their application to disaster risk management remains at its infancy and their potential to improve social capital, risk awareness, and preparedness require further attention.

There are many digital technologies relevant to disaster risk management. We zoom into four digital technologies: mobile positioning data, social media crowdsourcing, drones, and satellite imaging. Previous work in the BuildERS project indicates that mobile positioning data, social media crowdsourcing, drones, and satellite imaging have the greatest innovation potential for disaster risk management (Latvakoski et al., 2022). The identification of these as relevant technologies in disaster risk management is also supported, outside of the framework of the project, by the reviews recently carried out by Izumi et
al.(2019), Munawar et al (2022), and Vermiglio et al (2022). We therefore proceed by focusing on mobile positioning data, social media crowdsourcing, drones and satellite imaging. We present an overview of the selected digital technologies below.

### 2.1 Mobile positioning data

Mobile positioning data provides information on the location of a mobile device and its user (Raper et al., 2007). Two forms of mobile positioning data exist: *active*, when the system operator constantly tracks the mobile device, or *passive*, when the
90 system operator only tracks the mobile device while it is being used. The data encompass pseudonymized IDs, timestamps, phone activity, cell tower ID, and the national origin of the SIM card (Latvakoski et al., 2020; Võik et al., 2021). In a European context, the General Data Protection Regulation (GDPR) prohibits the use of real-time data if not anonymized. Mobile positioning data is often therefore provided with a minimum of 24-hour delay to guarantee the anonymity and privacy of the mobile device users (Bayardo & Agrawal, 2005; Lasko & Vinterbo, 2010; Terrovitis et al., 2008). Privacy is indeed a concern

when operating with this kind of data. For instance, privacy concerns over the sharing of real-time data are identified through the interviews that Bowser et al. (2017) conducted with project managers that handle this kind of data.

Mobile position data can be collected to mitigate disaster risk (Indriasari et al., 2017;  Munawar et al., 2022; Hirata et al., 2018; Paul et al., 2021). The ubiquity of smartphones across the globe has made it possible for authorities to use mobile positioning data to alert the population about an approaching hazard by sharing information through SMS messages and others means of communication (Grantz et al., 2020). The application designed by Leelawat et al.(2017) constitutes a good example of such a potential use: risk awareness is increased by providing the population with an assessment of the risk they are exposed to in the event of a tsunami based on their position. Mobile positioning data can also improve data collection, serving as an input when mapping disaster impact, drafting risk and vulnerability assessments, and monitoring mobility. The 2020 Covid-19 pandemic accelerated the use of mobile positioning data (Ekong et al., 2020; Ienca and Vayena, 2020; Santamaria et al., 2020), employed to assess the exposure to infection across the population (Grantz et al., 2020). The review performed by  Yabe et al. (2022) is an excellent source to understand and classify recent applications of such a technology in the domain of disaster risk management: access to mobile positioning data allows for an understanding of the displacement patterns in the affected population and for a study of the evacuation dynamics, with the possibility to predict post-disaster behaviours of future events based on the previous experiences, contributing to better approaches both in the response and in the preparation phases of the disaster risk management cycle; relocation patterns in the aftermath of a disaster have also been found to correlate with the amount of damage inflicted on the built environment, a condition that allows to proxy the damages estimation by observing relocation behaviours (Andrade et al., 2018); damage estimates and impacts on local businesses can also benefit from the availability of mobile positioning data (Yabe et al., 2020). The popularity of such an application cannot disguise the challenges that its application entails: the management of the data, which require discretion given their consequences for people's privacy and the difficulty in translating the analysis of the data into insights that can be easily understood by policymakers and hence turned into effective policies (Yabe et al., 2022).

## 2.2 Social media crowdsourcing

Social media crowdsourcing is the practice of contributing to the monitoring of disasters by reporting the presence of incidents related to the event through social media (Phengsuwan et al., 2021). Smartphones and social media use are heavily intertwined. Social media has gained popularity in disaster risk management due to its outreach potential (Reuter and Kaufhold, 2018). The possibility to establish two-way communication in which private citizens interact with official authorities allows for crowdsourcing (Besaleva & Weaver, 2016; Hernandez-Suarez et al., 2019; Kankanamge et al., 2019; Li et al., 2021; Ogie et al., 2019). There are two types of contribution: *active,* when social media users actively share information to support an initiative, and *passive,* when social media users share information independently and irrespectively of an initiative (Besaleva and Weaver, 2016). Typically, social media crowdsourcing generates large datasets that can be analysed using pre-trained high-level natural language processing software employing artificial intelligence (AI), machine learning, or blockchain technology (Latvakoski et al., 2020).

Social media crowdsourcing has a history of being deployed in disaster risk management, of which some notable examples include the 2010 Haiti Earthquake, 2014 North Stradbroke Island Bushfires (Australia), and 2015 Houston Flooding (USA)
(Kankanamge et al., 2019). Social media crowdsourcing allows the general public to get involved by reporting the conditions they find themselves in and their position: photos and videos can be geotagged once shared on social media, and ad-hoc apps can be downloaded to enable a two-way communication between first responders and those in need. Authorities can use this information to construct hazard maps and alert other citizens about the increasing risks if located in proximity (Zachreson et al., 2021). . Besides this flow of information exchange to construct almost-real time maps and inform citizens on the presence
of risks (Ogie et al., 2019), the amount of data collected through citizens engagement in social media can also be exploited to provide rapid assessment of the damage, either through direct observation of the messages shard online (Kryvasheyeu et al., 2016) or by applying a sentiment analysis that reveals the correlation between the sentiment level and the impact of the disaster (Li et al., 2021). The experience with past applications of such a technology has helped the literature to identify the challenges that may hinder an effective application: beside the need to constantly train and update the models used for data analysis and
interpretation, practitioners will have to design strategies that guarantee a constant, large and reliable source of data from the citizens. While they may appear as the easiest solution, monetary incentives risk undermining the altruistic reasons that push citizens to contribute to the application of this technology (Ogie et al., 2019).

**2.3 Drones**

There is a growing interest in unmanned aerial vehicles, of which drones represent the most popular example (Gomez and
Purdie, 2016). Drones are used in a vast range of sectors, such as warfare and agriculture, but also disaster risk management (Aydin, 2019). Significant improvements in miniaturization and computerization have enabled the production on a large scale of lighter, safer, and cheaper drones (Hall and Wahab, 2021).
The review carried out by Mohd Daud et al. (2022) highlights the operations in disaster risk management that can be aided by the use of drones. Beyond the standard use of drones to construct real-time maps of the areas affected by disasters such as
floods, landslide, wildfires and earthquakes with a rapidity and a cost-effectiveness that has often justified their adoption over other image-providing tools, drones have also found vast application in difficult-to-access areas. An alternative use is that of using drones to perform a rapid assessment of the damages to the built environment without having to put the personnel at risk. In this sense, drones are often sought after by practitioners as they increase they allow to perform some tasks in safe conditions (Wankmüller et al., 2021). Mohd Daud et al. (2022) also stress the relevant role that drones can play in search and
rescue operations: their review identifies several applications where the accessibility to geographical information on the position of the drone and the use of thermal cameras significantly increased the chances to find and rescue people that went missing during a disaster. Transportation of medical and emergency supplies during an emergency situation or in areas that would otherwise be difficult to reach is another task that can be performed with drones and that has contributed to the popularity of this tool (Rejeb et al., 2021). Further improvements are, however, needed in order to allow for a more efficient helicopter-

drone and drone-drone cooperation to reduce the risk of collision; and image and video quality will definitely benefit from future technological developments in image acquisitions (Wankmüller et al., 2021).

## 2.4 Satellite imaging

Satellite imaging is a  remote sensing technology taking pictures with an overhead perspective to detect patterns over large areas (Campbell and Wynne, 2011). Satellite imaging made its debut in the 1960s in the first real program for the acquisition

of imagery of Earth from space, the Landsat program, which was launched by the US government in 1972 (Hemati et al., 2021). The Integrated Global Observing Strategy (IGOS) initiative was launched in 1988 and provided satellite images to support efforts towards disaster risk management. Several initiatives followed, increasing the number of applications of satellite imaging to disaster risk management: the European "Copernicus" program and its Sentinel missions were deployed in 2014 and constitute, to date, the largest European initiative. Despite such a long history compared to other technologies,

satellite imaging has not found an early broad application in the domain of disaster risk management: the first 1-m resolution commercial satellite images were only made available thanks to the IKONOS satellite that was launched in 1999; the acquisition of images still takes around 48 hours after the event has occurred despite international efforts and collaborations to guarantee a shorter delivery (Le Cozannet et al., 2020). Current image quality and time of delivery already constitute a significant improvement compared to the standards of the past and despite these issues, satellite imaging has now emerged as

a standard technology in the field of disaster risk management thanks also to recent developments in AI and machine learning analysis of the images  (Dubovik et al., 2021; Goniewicz et al., 2020; Tellman et al., 2021; Wheeler & Karimi, 2020).
Le Cozannet et al. (2020) offers several points of view to understand and classify the venues for application of this technology. In particular, the authors argue, satellite imaging can be employed to aid in the prevention as well as in response to disasters. The benefits such technology bring in the prevention of the disaster are acting on those factors that determine the exposure to

disasters, such as the hazards themselves, as well as vulnerability and exposure. We here exemplify how satellite imaging can be useful across all these aspects. Reducing hazards often requires access to hazard maps, which are produced by observing the areas and the territories of interest. This makes satellite imaging particularly useful, such as in the case of understanding the characteristics around a volcanic area (Neri et al., 2013) or to estimate geological processes such as ground deformation without the need for in situ observations (Foumelis et al., 2016). Targeting vulnerability requires information to produce

vulnerability and fragility curves, such as buildings' shapes, width of streets and size of buildings (Menichini et al., 2022). Such data can be collected by direct observations but evidence from the recent literature shows the limit of such an approach and suggests that combining this with space observations guarantees better results (Geiß et al., 2014; Le Cozannet et al., 2018). Substantial improvements in exposure reduction can be achieved by avoiding exposing assets to the risk of hazards in first place or by relocating those that are now at risk and high-resolution satellite images can provide the level of details needed to

understand the dynamics in place (Tellman et al., 2021; Weichenthal et al., 2019). While future improvements in computational methods and data quality will make this technology more attractive (Teodoro and Duarte, 2022, p.10), current developments will unlikely be able to fulfill this promises immediately and satellite Earth observations will probably need to be paired with

alternative sources to accommodate for issues such as uneven temporal sampling (Frasson et al., 2019). The timing aspect appears to be particularly relevant in those cases where longer observations may be needed as the hazards hit: this is the case,

for instance, of floods, with their typical duration of few hours that can, at times, hardly be matched by satellites' capabilities (Almar et al., 2023).The accuracy of the measurements represents another aspects that highlights the limitations of this tool (Almar et al., 2023; Melet et al., 2020)

## 3. Materials and method

We apply a multicriteria analysis following the standard approach outlined in previous literature: i) criteria are identified, ii) a scale for the scoring options is defined, iii) criteria weights are produced, and iv) the final scores are computed using the weights from step iii) and the results from a survey (Bana e Costa et al., 2004; de Brito & Evers, 2016; Gamper et al., 2006). We perform a local sensitivity analysis where we apply different weighting schemes and different approaches for the aggregation of weights and scores into final figures.

### 3.1 Criteria identification

The criteria are identified by referring to the work of Morsut et al. (2021), as developed within the framework of the BuildERS project. The framework outlines the interlinkages between vulnerability, resilience, risk awareness, and social capital. The selection of the criteria aligns with the theoretical approach of the previous literature that confirms the interlinkages between these terms: Barua et al.(2020) on the connection between preparedness and vulnerability, Bixler et al. (2021) on the links
between social capital, and preparedness, Hanson-Easey et al.(2018) on the relationship between social capital and risk awareness and Liu et al.(2022) for a discussion on social capital and resilience .We include two additional categories to better reflect potential practical barriers that may emerge: feasibility and acceptability, with special attention towards regulatory frameworks, costs, and social acceptance (Barquet and Cumiskey, 2018; Georgieva, 2015).

Each of these five categories is then further developed into criteria. We identify the criteria in a one-hour workshop with project partners. The workshop is held online to allow project partners representing different European countries to participate. The participants represent different disciplines and professions all pertaining to the sphere of disaster risk management. In total, 9 people participate in the workshop and 15 criteria are identified (see Table **1** for an overview). We juxtapose the criteria with previous research on societal resilience to ensure their relevance (Carone et al., 2018; DFID, 1999; Hernantes et al., 2019;
The Rockerfeller Foundation, 2016).

| Category | Criterion |
|---|---|
| Risk awareness | $C_1$: It can strengthen risk and vulnerability assessments. |

| | | |
|---|---|---|
| | $C_2$: It can improve access to crisis information. | |
| | $C_3$: It can improve the quality of crisis information by making it more accurate, timely, or relevant. | |
| Social capital | $C_4$: It can minimize the risk of myopic thinking by strengthening trust and coordination between different organizations. | |
| | $C_5$: It can support citizen engagement in disaster risk management | |
| | $C_6$: It can enable collaboration and coordination of volunteers. | |
| Preparedness | $C_7$: It can improve data-collection, procedures, methods, and sharing. | |
| | $C_8$: It can contribute to the development of plans and strategies to manage crises. | |
| | $C_9$: It can improve emergency management exercises. | |
| Feasibility | $C_{10}$: It can be viewed as financially viable considering costs and revenues. | |
| | $C_{11}$: It is likely that there is access to necessary human, infrastructure, knowledge, and technical resources for implementing and maintaining the service. | |
| | $C_{12}$: It is likely to be supported by regulatory frameworks. | |
| Acceptability | $C_{13}$: It can meet local expectations in relation to the stated aims and services provided. | |
| | $C_{14}$: It receives local support from civil society and groups. | |
| | $C_{15}$: The perception of gains/needs is likely to outweigh perceptions of risk/threats. | |

**Table 1: Criteria**

## 3.2 Scoring options

We develop a survey to assess the digital technologies against the defined criteria. We first ask respondents about their field of expertise and the countries of operation. The respondents are thereafter presented with Likert scale questions outlining the 15 criteria listed in Table 1. We use a four-point Likert scale. We ask the respondents to express whether they "Strongly Agree", "Agree", "Disagree", or "Strongly Disagree" with the statement. We also add a fifth "I do not know" option. The same questions are asked for the four digital technologies to allow for comparison. The survey is launched online on October 5th, 2021, and closed on October 21st, 2021. A reminder is sent halfway to increase the response rate.

We invite respondents with expertise in disaster risk management or any of the targeted digital technologies to complete the survey. Respondents are contacted by email and asked to complete an online survey. In total, 118 responses are collected, but only 116 are used as two respondents failed to complete the survey. All respondents are anonymous. The respondents represent different geographical regions, including Europe (60%), Asia (15%), and North America (5%). The respondents represent different fields of expertise, including disaster risk management (n=79), remote sensing (n=36), drones (n=18), AI (n=18), social media crowdsourcing (n=16), mobile positioning (n=11), and Internet of Things (n=8). Note that the respondents can indicate more than one field as their domain of expertise.

We then convert the qualitative data into quantitative data to allow for the subsequent quantitative analysis. "Strongly Agree" is awarded four points and then decreasing values down to one for "Strongly Disagree". "I do not know" replies are dropped as they do not contain any insightful value for the analysis.

We thereafter proceed with creating a performance matrix. The entry values of the matrix for each of the innovation under each criterion are represented as the mean scores that are attributed to the innovation by the survey respondents. We assume that all criteria represent positive contributions to disaster risk management, hence a higher score indicates a better performance and a positive contribution.

### 3.3 Weighting of criteria

Weights are normalized values between 0 and 1 and add up to 1 when summed for all criteria within the same method. This may sometimes not be the case due to rounding. We apply a quantitative approach when weighting the criteria, using a multi-method framework which allows us to investigate if different methods will lead to different results. The following weighting methods are considered: the Criteria Importance Through Criteria Correlation (CRITIC) (Diakoulaki et al., 1995; Krishnan et al., 2021; Lai and Liao, 2021; Tuş and Adalı, 2019), the (Euclidean)-distance Criteria Importance Through Criteria Correlation (dCRITIC), the Standard Deviation method (SD), the Mean Weight method (MW) and the Entropy method (Deng et al., 2000; Xu, 2004), all of which represent common approaches in the literature (Diakoulaki et al., 1995; Xu, 2004; Deng et al., 2000; Tuş and Adalı, 2019; Lai and Liao, 2021; Krishnan et al., 2021; Sun et al., 2020; Odu, 2019). An overview is provided Table 2. We only provide a brief description of them while the reader is invited to refer to the previous literature for an analytical analysis.

| Name | Description |
|---|---|
| CRITIC | CRITIC is inspired by the analysis proposed by Diakolulaki et al. (1995), which is based on a correlation analysis of the criteria. The standard deviation method is meant to capture the contrast intensity. Diakolulaki et al. (1995) add a correlation study on top of that to account for emerging conflicts among the criteria. |
| dCRITIC | dCRITIC, as proposed by Krishnan et al. (2021), represents a small variation in the way correlations, variances, and standard deviations are computed in comparison to the CRITIC method. |
| Standard deviation | The standard deviation (SD) method presents several similarities to the Entropy method in the way the weights are computed. The method attributes low weights to the criteria that have similar scores across all the options. The idea behind such a result is straightforward and intuitive: criteria that do not vary across the options do not provide meaningful results and should therefore be assigned a low weight (Zardari et al., 2015). |

| | |
|---|---|
| Mean weight | The mean weight method does not present any computational complexities as every criterion is assigned the same weight, which is just the ratio of 1 over the number of criteria. This method makes no use of the information contained in the performance matrix and just assigns the same weight to all criteria. |
| Entropy | The entropy method entails a certain degree of uncertainty in the information matrix that is accounted for by making use of probability theory. It suggests that spread and broad distributions contain more uncertainty than the sharply peaked ones (Zardari et al., 2015). |

**Table 2: Weighting methods**

We delve into the literature to select the weighting methods. A review by Krishnan et al. (2021) finds the Entropy and the CRITIC methods to be the most frequently used methods in the literature. The authors also suggest that the dCRITIC methodology requires more attention as it may deliver results that differ from those of the CRITIC method. We add the MW method, as it is a common method in the vulnerability and resilience literature (Tate, 2012). We also add the SD method, as it makes an interesting case for comparison due to its similarities with the CRITIC and dCRITIC methods (Krishnan et al.,
2021).

### 3.4 Aggregation

The weights obtained through the different weighting methods and the mean scores obtained through the survey are used to rank the four alternatives following a Weighted Sum Approach (WSA):

$$Q_i = \sum_{j=1}^{n} x_{ij}^* w_j$$

(1)


where $x_{ij}^*$ is the average survey score of technology $i$ under criterion $j$ and $w_j$ is the weight of criterion $j$. In order to provide additional sensibility to the analysis, we aggregate the results through the Weighted Product Approach (WPA) too, another methodology that is often employed in the literature (Odu, 2019):

$$Q_i = \prod_{j=1}^{n} (x_{ij}^*)^{w_j}$$

(2)

**4. Results**

In this section, we first present responders' opinions as we collected them through the survey. We aggregated these into mean score for each technology under each criterion. We then proceed with applying the different weighting methods to investigate their impacts on the findings. Lastly, we present the aggregated findings.

**4.1 Survey responses**

This section presents the survey responses. An overview is provided in Figure **1**. The x-axis presents the criteria, ordered and numbered according to Table 1. The y-axis shows the mean score for each innovation under the corresponding criterion. The range of the y-axis has been reduced to better appreciate the difference in scores that each technology was awarded by the participants to the survey.

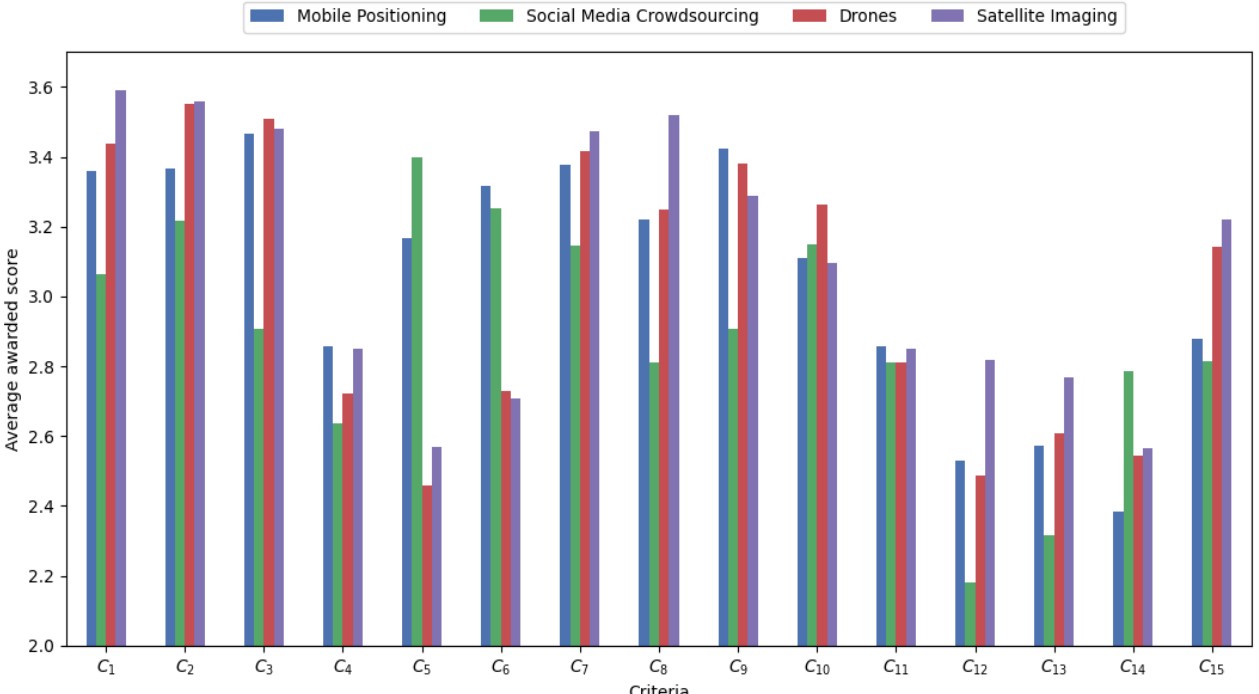

**Figure 1: Questionnaire results (average scores)**

Satellite imaging has the highest mean score for Criterion 1 (*"It can strengthen risk and vulnerability assessments"*) and Criterion 2 ("*It can improve access to crisis information"*). Drones score the highest in Criterion 3 (*"It can improve the quality of crisis information by making it more accurate, timely, or relevant"*). Social media crowdsourcing receives the lowest mean score in all criteria related to risk awareness, Criteria 1, 2, and 3.

Looking at social capital, mobile positioning data holds the highest mean score for Criterion 4 (*"It can minimize the risk of myopic thinking by strengthening trust and coordination between different organizations"*) and Criterion 6 (*"It can enable*

*collaboration and coordination of volunteers"*). Social media crowdsourcing has the highest mean score for Criterion 5 (*"It can support citizen engagement in disaster risk management"*). Both drones and satellite imaging score low in all criteria related to social capital.

When it comes to preparedness, satellite imaging scores the highest in Criterion 7 (*"It can improve data-collection, procedures, methods, and sharing"*) and Criterion 8 (*"It can contribute to the development of plans and strategies to manage crises"*). Drones have the highest mean score in Criterion 9 (*"It can improve emergency management exercises"*). Social media crowdsourcing receives the lowest mean score across all criteria.

In terms of feasibility, all digital technologies receive a high mean score for Criterion 10 (*"It can be viewed as financially viable considering costs and revenues"*), of which drones score the highest. Mobile positioning data have the highest mean score in Criterion 11 (*"It is likely that there is access to necessary human, infrastructure, knowledge, and technical resources for implementing and maintaining the service"*). All digital technologies score low in Criterion 12 (*"It is likely to be supported by regulatory frameworks"*). Social media crowdsourcing scores the lowest in Criterion 12.

In the final category, acceptability, most digital technologies scored low across all criteria. Mobile positioning data received the lowest overall score, whereas satellite imaging received the highest overall score. Satellite imaging has the highest mean score in Criterion 13 (*"It can meet local expectations in relation to the stated aims and services provided"*), and social media crowdsourcing has the highest mean score in Criterion 14 (*"It receives local support from civil society and groups"*). Satellite imaging and drones received a high mean score in Criterion 15 (*"The perception of gains/needs is likely to outweigh perceptions of risk/threats"*).

## 4.2 Weighting methods

This section presents the results from the weighting assessment. The weights are reported in Figure 2. The x-axis presents the 15 criteria. Please refer to Table 1 for an explanation of what they stand for. The MW method is not included in Figure **2**, as it assigns all criteria the same weight by construction. The weighting methods produce different rankings of criteria and some of these differences are substantial.

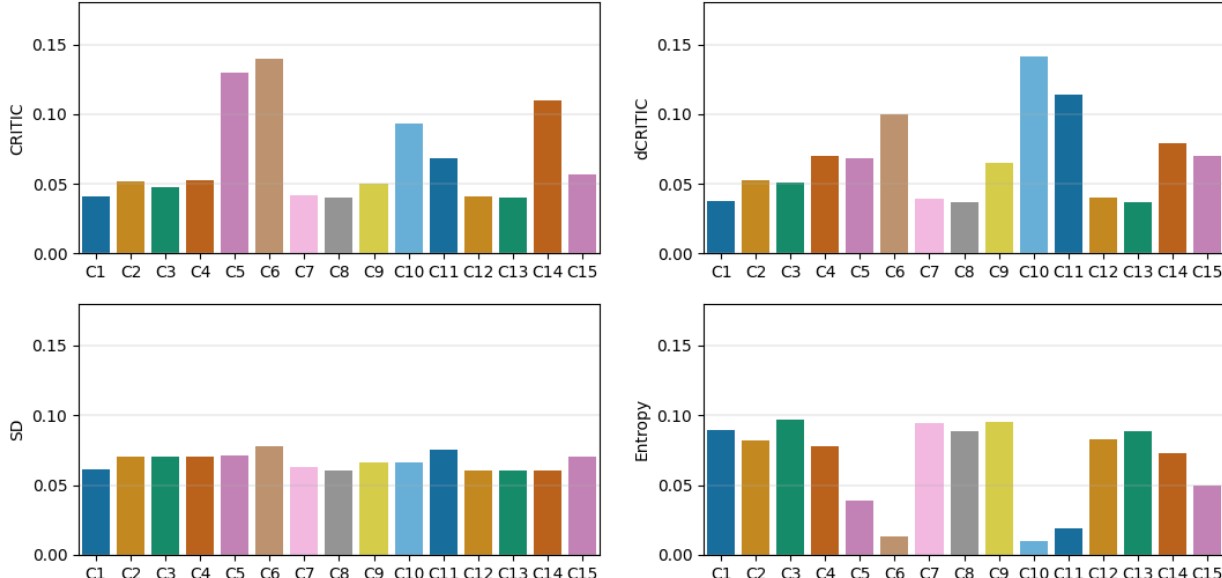


**Figure 2: Methods' weights**

Weights' ranges are large in both the CRITIC and the dCRITIC methods. Figure **2**Under the CRITIC method, the largest weight is about 13.5% and the smallest weight is about 4%. Similarly, under the dCRITIC method, the largest weight is about 14% and the smallest weight is about 3.8%. That is, for both methods the largest weight is more than 3 times the smallest

weight. Technologies that score high in the criteria with the largest weights will rank higher. Criterion 6 ("*It can enable collaboration and coordination of volunteers.*") is assigned the largest weight under the CRITIC method, whilst Criterion 8 ("*It can contribute to the development of plans and strategies to manage crises.*") receives the lowest weight. The distance correlation correction that is introduced by the dCRITIC method seems to bring some changes, though not substantial, when compared to the original CRITIC method: Criterion 10 ("*It can be viewed as financially viable considering costs and*

*revenues.*") receives the largest weight, while Criterion 6 still ranks among the top three. Criterion 13 ("*It can meet local expectations in relation to the stated aims and services provided.*") receives the lowest weight under the dCRITIC method and it ranks 14[th] under the CRITIC method. To conclude, some changes are indeed produced by computing a distance correlation instead of the standard Pearson correlation. The changes are, however, minor at the top and bottom of the weights rank.

The SD method and the CRITIC method return similar results in relative terms, i.e. which weights are assigned the largest

weights, but the values of these are different. The CRITIC method uses the standard deviations within each criterion to compute the correlation matrix and to correct the values of the final weights and this could explain the similarity. The SD method assigns the highest weight value to Criterion 6 ("*It can enable collaboration and coordination of volunteers*"), just like the CRITIC method. The smallest weight in the SD method is assigned to Criterion 12 ("*It is likely to be supported by regulatory frameworks*"), which has the third smallest weight under the CRITIC method. That is, in relative terms, the two methods seem

to assign large weights to the same criteria, but the actual weights determined by the SD method are much closer to each other

than those observed under the CRITIC one. In fact, we find the SD method to be the one, after the MW method, to return the smallest range of weights.

SD and MW methods lead to similar distribution of the weights: both methods return a narrow range. The weights defined by the SD method are narrowly distributed: the maximum weight is about 7.8% and the minimum is about 5.9%, with a difference between the largest and the smallest weight of about 1.9 percentage points. This can be seen in Figure 2, where although the bars have different heights, overall, the graph looks "flat", with no criterion being assigned a weight that is much larger than the other criteria and hence no bar being significantly taller than the others. The difference between the largest and the smallest weight in the case of the MW method is 0 as all the criteria are awarded the same weight by construction. Each criterion is assigned a weight of 0.066 or 6.66% under the Mean Weight method. A barplot of the MW's weights would show a set of 15 identical bars. If all the weights are similar, as in the case of the results produced by the SD or MW methods, the innovation with the highest average performance score across all criteria will rank higher, which can be supported by statistical claims. By exploiting the narrow range of the SD and of the MW methods one could be able to predict, with a significant degree of confidence, which technology will rank first under these methods by simply looking at the survey results as presented in Figure 1: even without resorting to statistical analysis, it is evident that satellite imaging scores the highest under most of the criteria (7 out of 15), is expected to have a high average score and will probably rank high using the weights determined by the SD and MW methods.

The Entropy method returns weights that are different from those found in other methods. A large range of different weights values is observed under this method too as in the CRITIC and dCRITIC methods, but the criteria awarded the largest weights are different from those of other methods. Five of the criteria alone, Criterion 15 (*"The perception of gains/needs is likely to outweigh perceptions of risk/threats"*), Criterion 1 (*"It can strengthen risk and vulnerability assessments"*), Criterion 13 (*"It can meet local expectations in relation to the stated aims and services provided"*), Criterion 12 (*"It is likely to be supported by regulatory frameworks"*) and Criterion 8 (*"It can contribute to the development of plans and strategies to manage crises"*), account for about 40% of the total weight and satellite imaging scores first under all of these criteria. It is reasonable to expect this innovation to score high under this method. As for the CRITIC and for the dCRITIC methods we observe a large distribution of the weights awarded to the criteria. It is interesting to note that some of these receive similar weights under the three methods, CRITIC, dCRITIC and Entropy, such as Criterion 14 ("*It receives local support from civil society and groups*") and Criterion 15 ("*The perception of gains/needs is likely to outweigh perceptions of risk/threats*"). However, a striking aspect that should already be observed from Figure 2 is that the most relevant criteria under the CRITIC and dCRITIC methods, i.e. those that are awarded the largest weights, are assigned the smallest weights under the Entropy method. Just by looking at these preliminary results it is reasonable to expect large differences in the final rankings among these methods.

The Spearman correlation matrix in Table 3 provides additional insights for a comparison of the weights that are produced by the different methods. The correlation between the entropy method and the CRITIC and dCRITIC methods is negative and

large, meaning that the ranking order may be really inverted and that the weights vary significantly, confirming what was observed in Figure 2. As expected, the dCRITIC and CRITIC methods have a high positive correlation value as the way in which the correlation matrix is computed constitutes the only difference between the two. No correlation can be computed between the MW method and any of the other methods as there is no variation in the assigned weights of the former. That is, the variance of the MW method is 0. A null denominator in the correlation formula impedes any computation.

| | *Entropy* | *MW* | *dCRITIC* | *SD* | *CRITIC* |
|---|---|---|---|---|---|
| *Entropy* | 1 | - | -0.79 | -0.45 | -0.76 |
| *MW* | - | - | - | - | - |
| *dCRITIC* | -0.79 | - | 1 | 0.575 | 0.9 |
| *SD* | -0.45 | - | 0.575 | 1 | 0.65 |
| *CRITIC* | -0.76 | - | 0.9 | 0.65 | 1 |

**Table 3: Spearman correlation matrix**

### 4.3 Aggregation

Figure 3Figure 3 offers an overview of how the four digital technologies rank under the different weighting methods. Only results from the Weighted Sum Approach are presented as the results produced by the Weighted Product Approach are identical. The rankings are produced according to the final score that is computed for every technology: we use the weights produced by the methods, presented in Figure 2, and the average survey scores in Figure 1 by feeding these values into Equation 1 and Equation 2, presented in the Material and Methods section. We find here a confirmation of the correlations that were produced in Table 3: the high negative correlation between the Entropy method and the CRITIC and dCRITIC methods results in a complete reversal of the ranking order. The Entropy method returns results that are, at the top and bottom, the same as those returned by the SD and MW methods: satellite imaging still ranks as the most preferred innovation, and social media crowdsourcing as the least.

As expected from the computation of the Spearman correlation matrix, the CRITIC, dCRITIC, and SD methods are the ones that produce the closest results. Across all weighting methods, with the sole exception of CRITIC, social media crowdsourcing seems to be the least preferred option among respondents. Satellite imaging ranks as the best innovation in three out of five methods (SD, MW, and Entropy) with mobile positioning data ranking first in the remaining two. Overall, the results suggest that satellite imaging is the most preferred innovation. It is evident, however, that the results are very sensible to the methodology used. The best innovation according to the entropy method, satellite imaging, ranks third under the CRITIC method. And the opposite is true for mobile data positioning, which ranks first under the CRITIC method and the dCRITIC method but then falls to the third place under the entropy method.

As we had been able to anticipate, satellite imaging is the technology that ranks first under the SD and the MW method. We attribute this, again, to the small range of weights produced by these two methods. It is also interesting to note that these two are the only methods where the ranking orders are exactly identical.

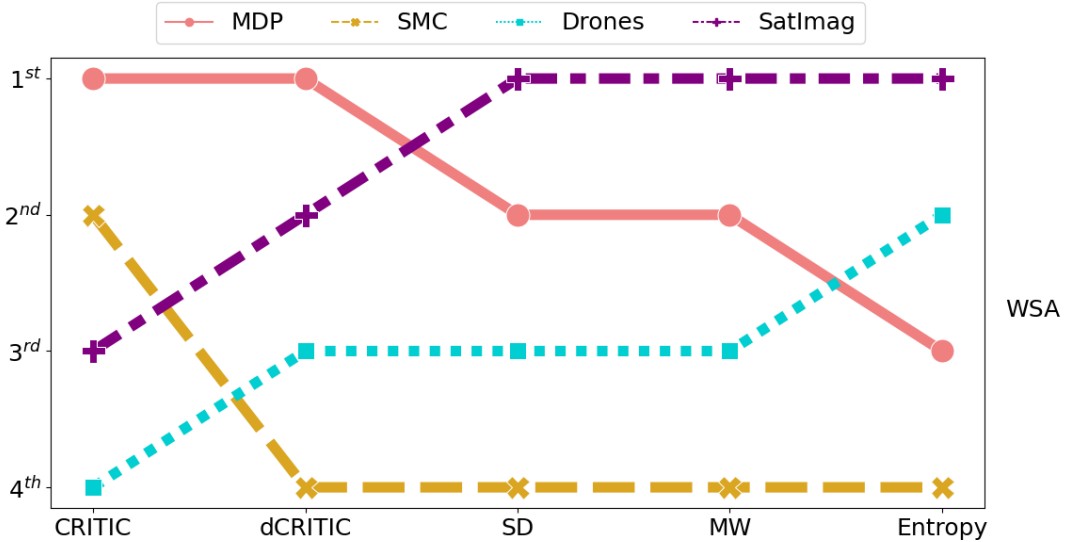

**Figure 3: Final ranking of the technologies**

## 5 Discussion

Below we first discuss the findings in relation to disaster risk management and societal resilience, with a focus on their impact on social capital, risk awareness, and disaster preparedness. We thereafter delve into the weighting methods and make some recommendations for scholars assessing digital technologies for disaster risk management.

### 5.1 Digital technologies for enhancing societal resilience

The different weighting methods produce different results, which makes it difficult to draw conclusions about what digital technologies can improve social capital, risk awareness, and disaster preparedness. In general, the findings show positive or neutral perceptions of the digital technologies under investigation. When juxtaposing the digital technologies, we find some emerging patterns that indicate a preference of some technologies over others.

Social media crowdsourcing received the lowest score in the survey in all criteria except Criterion 5 (*"It can support citizen engagement in disaster risk management"*), Criterion 10 (*"It can be viewed as financially viable considering costs and revenues"*), and Criterion 14 (*"It receives local support from civil society and groups"*). We attribute the poor ranking to the interdependence between social capital, risk awareness, and disaster preparedness in which a negative performance in one of them may reinforce negative performance in the others. It creates a viscous circle of cascades and unintended consequences.

In the case of social media crowdsourcing, scholars find that it may generate data inaccuracies due to mis/disinformation and social biases (Fang et al., 2019; Nguyen et al., 2019). Data inaccuracies may have a negative impact on preparedness if integrated into risk and vulnerability assessments, contingency plans, and emergency exercises. Felice and Iessi (2019) note that inaccurate data generated from social media crowdsourcing may exacerbate disaster impacts. In line with previous research (Han et al., 2011; Nicholls and Picou, 2013), we believe that ineffective disaster response can erode social capital and trust as it might cause a loss in confidence in disaster risk management authorities.

Regardless of the weighting method, satellite imaging and drones yield similar benefits for risk awareness and disaster preparedness including improvements in data collection; better input to risk and vulnerability assessments; support to emergency exercises and contingency plans; and improvements in quality and access to crisis information. Given these similarities, satellite imaging and drones may be considered as alternatives i.e., investments should be made in one or the other but not both as they yield similar benefits. However, in line with previous research (Sajjad and Kumar, 2019; Gray et al., 2018; Kucharczyk and Hugenholtz, 2021), we argue that satellite imaging and drones can complement each other when applied in disaster risk management. Satellites and drones can support different aims, and offset different challenges as regard to accessibility and feasibility. As noted by Bansod et al. (2017), our findings indicates that satellite imaging and drones differs in terms of compliance with regulatory frameworks; financial viability; and social acceptance. Furthermore, satellites and drones offer imagery at different scales and therefore have different purposes and attributes. In disaster risk management, drones are used on a local scale and case-by-case basis while satellite imaging is more standardized.

Practical barriers seem to pertain to all digital technologies regardless of the weighting method, which impede their integration in disaster risk management despite their potential benefits for social capital, risk awareness, and disaster preparedness. Noncompliance with regulatory frameworks, inadequate access to resources, and limited social acceptability apply across all digital technologies. In line with previous research (Gambino and Tuzzolino, 2022; Bu-Pasha, 2018; Tsiamis et al., 2019; Harrison and Johnson, 2019; Santos and Rapp, 2019), this indicates a gap between technological advancements and institutional contexts. Technological development seems to accelerate faster than regulatory frameworks and capacity development. We see a need for going beyond technological solutions and consider the practical constraints faced by actors in disaster risk management, in order to strengthen social capital, risk awareness, disaster preparedness, and societal resilience.

We collected data from respondents representing various regions with differences in disaster exposures and social conditions. Vulnerabilities are represented differently depending on context hence requiring different types of disaster risk reduction measures. For example, satellite based remote sensing cannot operate in cloudy conditions, whereas drones can (Emilien et al., 2021). This study does not capture these contextual and situational differences. We, therefore, recommend future research to investigate social capital, risk awareness, and disaster preparedness in depth and in context, in order to make recommendations for policymakers and practitioners.

## 5.2 Reflections on the weighting of the criteria

We find that the choice of weighting method has a significant impact on the final ranking of the alternatives, while the way in which survey results and weights are aggregated to produce the final scores does not affect the ranking order. The final rankings produced by the methods are the results of the underlying assumptions on how to better treat the uncertainties and the distributions of stakeholders' opinions. We, therefore, refrain from recommending one method over another. Instead, we suggest applying different weighting methods to observe what outputs are produced under different assumptions. This can

mitigate potential uncertainties and biases. Using different weighting methods can increase robustness, reduce uncertainty, and improve credibility of the results.

Our findings suggest that, when selecting the methods for evaluating alternatives, methods that share similar underlying assumptions and compute scores in a similar way should be avoided as the additional information they provide is not substantial and risk reinforcing the decision bias. The combination of methods with diverging theoretical assumptions can better recognize

the uncertainty found in the dataset and account for unseen patterns. In our case, the CRITIC and dCRITIC methods yielded similar results and are hence insufficient on their own, as they both rely on an approach that is based on a correlation analysis among the criteria. The similarity of results under these two methods can be explained by the distance correction that the dCRITIC method adds to the CRITIC method and that has little effect on the computation of the results. It can be said that these two methods belong to the same family, as both stress the analysis of the correlation between the criteria as their main

feature. Both methods thus build on similar assumptions. Because of this common trait, they may award similar weights to the same criteria leading then to similar results. We observed that the only two methods that produced the same exact ranking preferences of the technologies are the SD method and the MW method. We argue that this result is due more to a selection choice by the researchers than to the similarity in the computational steps. The MW method returns a null range of weights by construction: all criteria are assigned the same weight. The fact that we observe similar results from the SD method, we argue,

is instead linked to the selection of a 4-points Likert scale. The selection of such a small scale returns a mean that is significantly close to the median. To provide an example, Mobile Positioning Data has an average score of 3.36 under Criterion 1 ("*It can strengthen risk and vulnerability assessments*") and the distribution of scores for this technology under this criterion does not differ much from this mean: 51 of the respondents awarded 3 points and 50 awarded 4 points. The narrative does not change for the same technology under Criterion 2 ("*It can improve access to crisis information*"), where the mean score is 3.37 and

53 respondents awarded 3 points and 49 respondents awarded 4 points. The argument we are stressing is that with these distributions of results from the survey, all the criteria present similar, small, standard deviations. As the SD method awards each criterion a weight that is expressed as the ratio between that criterion's standard deviation and the sum of all standard deviations across all criteria, it is clear that the weights will have close values and will then show a distribution that is similar to the one obtained with the MW method. The distribution of survey results, centered closely around the mean and therefore

all very close to each other, is the reason, we argue, why the SD method returns weights that have similar value, the "flat

distribution" observed in Figure 2. A larger Likert scale, i.e. with more scoring options, could potentially prevent such a result and return a different range of weights.

Additional methods that consider alternative weight specifications and assumptions should be included to provide a robust analysis of the performance. In this case, we rely both on the Entropy method to provide additional insights as it makes use of probability theory instead of correlation or standard deviation analysis, and on the mean weight method, which makes no assumption regarding the uncertainties in stakeholders' opinions by assigning every criterion the same weight. We show that the introduction of a method that builds on a different set of assumptions, the Entropy method, may completely change the results of the analysis.

## 5.3 Limitations

The geographical distribution of the responses collected through the survey should stand as a caveat against the external validity of the results presented here. The way local governments, communities, enterprises and other local actors interact generates a dynamic that makes every application extremely case-specific (Maskrey, 2011).Moreover, the adoption and the application of certain digital technologies in areas that are already characterized by uneven distribution of vulnerability and inequalities. The case holds both for the Global North, as shown in Wang et al.(2019), where social vulnerability of certain communities has been exacerbated by the use of social networking sites for information exchange during responses to Hurricane Sandy, and for the Global South as well, as exampled in the case of the 2015 earthquake that hit Nepal: technological innovation in disaster management were introduced in context of deep social and digital inequalities, benefitting mostly those less at risk (Mulder, 2020).

## 6. Conclusion

Our study makes a contribution to the literature on disaster risk management by evaluating digital technologies for improving social capital, risk awareness, and disaster preparedness. We applied a multicriteria analysis that incorporated different weighting methods in order to test uncertainties in the relative degree of importance of social capital, risk awareness, and disaster preparedness for societal resilience. The conclusions of our analysis, we hope, will benefit the academic community and practitioners as well. For the former, the warnings we raised on the implications of the choice of the model to aggregate stakeholders' opinions may raise awareness among researchers working with similar methods, even in a different field; for the latter, the conclusions of the analysis inform practitioners on the suitability of adoption one or more of the tools to achieve their goals of increasing awareness, social resilience and disaster-responsiveness.

To overcome some of our shortcomings, we recommend that future research to engage in comparative case studies and provide a contextual assessment of how digital technologies can improve social capital, risk awareness, and disaster preparedness. We suggest looking into an array of empirical contexts to test their application in various geographical areas and situations. Future research can also explore potential negative spill over effects that cascade across social capital, risk awareness, and disaster

preparedness when implementing the digital technologies in practice. We argue that the size of the scoring scale too may have a significant impact on the ranking of preferences in a multicriteria assessment. We therefore encourage further research on this topic to increase the awareness around this issue and prevent selection biases.

To conclude, our study shows the potential of digital technologies in strengthening social capital, risk awareness, and disaster preparedness. We highlight the importance of using rigorous methodologies when assessing different innovative solutions for disaster risk management. Sensitivity analysis by comparing different weighting methods and testing different assumptions on how to treat uncertainties from survey results can enhance the communication of findings and improve confidence in the reliability of the results. The number of choices that a researcher makes is high, from the selection of a scoring scale to that of

the weighting models, and this increases the likelihood of uncertain results that would not be confirmed by an alternative approach. Different assumptions and different approaches should always be considered as a mean to increase the robustness of the results. Above all, this can improve decision making and societal resilience and in the long run protect human lives and economic assets.

**Authors contribution**

**TP –** conceptualization, formal analysis, data curation, methodology, visualization, writing – original draft preparation, writing – review and editing

**ME –** conceptualization, formal analysis, methodology, writing – original draft preparation, writing – review and editing

**KB –** funding acquisition, supervision, writing – review and editing

**Code/Data availability**

The datasets generated during and/or analyzed during the current study are available from the corresponding author.

**Competing interests**

The authors declare no competing interests.

**Funding statement**

This work was supported by the Horizon 2020 BuildERS project, grant agreement No 833496.

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
