# Peer review of "An assessment of potential improvements in social capital, risk awareness, and preparedness from digital technologies"

_Natural Hazards and Earth System Sciences, 2023_

## Author Response (AR1)

We would like to express our gratitude and thank the reviewers for a fruitful discussion that we are sure will benefit the quality of the manuscript. Please find below our point-to-point replies to the comments with a description on how these were incorporated in the text.

| Reviewer #1 | |
|---|---|
| Social capital is not a usual concept in geosciences. In the introduction, the authors should define this concept using references. | We have added this definition in the introduction: "features of social organizations, such as networks, norms and trust that facilitate action and cooperation for mutual benefit" (Putnam, 1993, p. 35) (Line 51) |
| The following questions require answers: "How are these digital technologies used currently (what is the state-of-the-art)?", and "How are these digital technologies can be implemented effectively?" A systematic review can contribute to these answers. | Each of the sections introducing the digital technologies has now been expanded to present state-of-the-art applications and other considerations around them. We acknowledge that the discussion may have been deeper but a systematic review on these technologies is not the main scope of the paper and we also tried to keep everything within a reasonable word limit. Nevertheless, we made a review and added the following text:

[revised manuscript text omitted]

| | |
|---|---|
| The analysis and content reflect a North Global perspective. I suggest to mentioned in a specific section the limitations of the research. Some vulnerable groups can become more vulnerable in South Global from the application of digital technologies in the only way. Besides this, inequality in undeveloped countries or developing countries can hamper the wide and fair application of digital technologies. The sentence in line 37: "… the end improve societal resilience among the most vulnerable segments of the population" requires attention | Lines 76-78 presents the scope of the project in line with the goals of the project "the project aims to improve social capital, risk awareness, and preparedness among the most vulnerable segments of the European population …".

In order to stress the risks that come from digital technologies application in the Global South we have added a section "5.3 Limitations" that read as follows:
"The geographical distribution of the responses collected through the survey should stand as a caveat against the external validity of the results presented here. The way local governments, communities, enterprises and other local actors interact generates a dynamic that makes every application extremely case-specific (Maskrey, 2011).Moreover, the adoption and the application of certain digital technologies in areas that are already characterized by uneven distribution of vulnerability and inequalities. The case holds both for the Global North, as shown in Wang et al.(2019), where social vulnerability of certain communities has been exacerbated by the use of social networking sites for information exchange during responses to Hurricane Sandy, and for the Global South as well, as exampled in the case of the 2015 earthquake that hit Nepal: technological innovation in disaster management were introduced in context of deep social and digital inequalities, benefitting mostly those less at risk (Mulder, 2020). " (Lines 500 – 509) |
| More discussion and results analysis are required, such as, whether there is a relation between the knowledge area or profession, or country and the weights. Cluster analysis can be used. | While we agree that it could be interesting to investigate any such relationship, the way the survey was designed does not allow for such a possibility. The country of operation and the area of expertise were provided as open questions. Many respondents indicated multiple countries, multiple regions or entire continents as their geography of operation. Mapping these results to unique values may risk invalidating the actual answers provided by the responders. The same applied to the area of expertise, where many indicated multiple technologies or fields of study. We are ready to share the data with anyone upon request. |
| **Reviewer #2** | |
| Author/s need to clarify what are digital technologies and why they decide to focus on a specific group (i.e., mobile positioning data, social media crowdsourcing, drones, and satellite imaging.). For an in-depth analysis of the role of digital technologies in government please read and cite: Barcevičius, E., Cibaitė, G., Codagnone, C., Gineikytė, V., Klimavičiūtė, L., Liva, G., ... & Vanini, I. (2019). Exploring Digital | The selection of the technologies has been done in accordance with the scope of the Horizon2020 BuildERS project this manuscript contributes to. This is mentioned in lines 90-92: "Previous work in the BuildERS project indicates that mobile positioning data, social media crowdsourcing, drones, and satellite imaging have the greatest innovation potential for disaster risk management (Latvakoski et al., 2022). " with a reference to:

Latvakoski, J., Öörni, R., Lusikka, T., & Keränen, J. (2022). Evaluation of emerging technological opportunities for improving risk awareness and resilience of vulnerable people |

| | |
|---|---|
| Government transformation in the EU. | in disaster*s. International Journal of Disaster Risk Reduction, 80,* 103173.  https://doi.org/10.1016/j.ijdrr.2022.103173

We further strengthen our focus by relying on examples from the previous literature and highlighting that these constitute standard examples of digital technologies in disaster risk management: "The identification of these as relevant technologies in disaster risk management is also supported, outside of the framework of the project, by the reviews recently carried out by Izumi et al.(2019) Munawar et al (2022) and Vermiglio et al (2022), We therefore proceed by focusing on mobile positioning data, social media crowdsourcing, drones and satellite imaging." (Lines 92-95) |
| For a better understanding of the importance and the different role of DG in disaster situations, please read and cite: Vermiglio, C., Noto, G., Rodríguez Bolívar, M. P., & Zarone, V. (2022). Disaster management and emerging technologies: a performance-based perspective. Meditari Accountancy Research. | The paper has been included in the section "2 Digital technologies in disaster risk management" to provide further evidence on the relevance of digital technologies in the different phases (Line 84). Given its relevance for the identification of relevant technologies in disaster risk management, we have also cited it to justify the list of the technologies we focus on (Line 94) |
| A further theoretical issue of the paper is the  lack of clear explanation regarding "social capital" and "risk" and "resilience" concepts which are pivotal for the theoretical background of the paper. On this regard, I suggest to broaden the explanation considering the following papers:

ALDRICH D., MEYER M.A., (2015) Social Capital and Community Resilience. American Behavioral Scientist 2015, Vol. 59(2) 254–269;

ALEXANDER, D.E., 2013. Resilience and disaster risk reduction: an etymological journey. Natural Hazards and Earth System Sciences, 1, pp.1257– 1284.

CAPANO G., WOO J.J. (2016), Resilience and robustness in | Thank you for sharing a list of literature. We have added two paragraphs to the introduction to define social capital, risk, and resilience. We have also added additional references to further strengthen our links to the literature.

The first three paragraphs (Lines 34 – 61) in the introduction are rephrased as follows:

"The Sendai Framework for Disaster Risk Reduction (UNDRR, 2015) calls for investments in digital technologies and tools to enhance societal resilience. Recent developments in digital technologies and tools offer emerging opportunities for managing disaster risk, i.e., the potential for loss or damages determined by the function of hazard, exposure, and vulnerability (Disaster risk, 2023). More specifically, digital technologies and tools hold significant potential in strengthening social capital, risk awareness, disaster preparedness, and, in the end, societal resilience (Latvakoski et al., 2022).
Many scientific fields adopt the concept of resilience (Alexander, 2013), including ecology (Holling, 1973), psychology (Garmezy et al., 1984), and disaster research (Manyena, 2006). As a consequence, resilience is subject to diverse definitions and conceptualizations (see for example IPCC, 2014; Johansen et al., 2017; Joseph, 2018; Manyena, 2006; Morsut et al., 2021; UNDRR, 2015; Zhou et al., 2010). Some researchers suggests that resilience refers to the ability of a system to bounce back |

policy design: a critical appraisal. Policy Science. Springer.

MANYENA, S.B., 2006. The concept of resilience revisited. Disasters, 30(4), pp.433–50.

DUFTY, N. (2012). Using social media to build community disaster resilience. Australian Journal of Emergency Management, 27(1), 40–45

JURGENS M., HELSLOOT I., (2018), The effect of social media on the dynamics of (self) resilience during disasters: A literature review. Journal of Contingencies and Crisis Management, 79-88. John Wiley & Sons.

to its equilibrium (Capano and Woo, 2017; Jurgens and Helsloot, 2018). Other researchers, however, denotes the bounce back metaphor as it fails to capture changes in the social fabric that occur in the wake of a disaster (Dufty, 2012). Accordingly, resilience refers to the ability of a system to bounce forward to a new normal i.e., anticipate, recognize, adapt to and learn from societal disruptions and disasters (Becker, 2014).
There is a plethora of factors that enable or constrain resilience (Jordan and Javernick-Will, 2012). In disaster research, social capital has emerged as a critical determinant of resilience (Kerr, 2018). Social capital refers to "features of social organizations, such as networks, norms and trust that facilitate action and cooperation for mutual benefit" (Putnam, 1993, p.35).
Greater levels of social capital within a community are linked to higher levels of disaster preparedness and risk awareness (Brunie, 2007; Hausman et al., 2007; Morsut et al., 2021). The nexus between social capital, risk awareness, and disaster preparedness can improve and facilitate collaboration; provide social safety nets; strengthen communication and information-sharing; speed up response and recovery efforts; and in the end improve resilience among the most vulnerable segments of the population (Aldrich and Meyer, 2015)."
* * *
The methodology is interesting and well grounded, although author/s doesn't clarify the reasons that justify the adoption of the selected criteria.

We are now providing in Lines 227-231 references to link our selection of criteria to the findings of the previous literature:

"
The selection of the criteria aligns with the theoretical approach of the previous literature that confirms the interlinkages between these terms: Barua et al.(2020) on the connection between preparedness and vulnerability, Bixler et al. (2021) on the links between social capital, and preparedness, Hanson-Easey et al.(2018) on the relationship between social capital and risk awareness and Liu et al.(2022) for a discussion on social capital and resilience"

And Lines 238-239:
"We juxtapose the criteria with previous research on societal resilience to ensure their relevance (Carone et al., 2018; DFID, 1999; Hernantes et al., 2019; The Rockerfeller Foundation, 2016)"
* * *
Practical and managerial implications of the study are not fully explained. Who can benefit from the results of this study? Managers? Policy makers? Practitioners? Academic community? You must clarify the target and create a strong linkage among theories, methodology and findings.

Potential beneficiaries of the results are now specified in the conclusion and reasons are presented on why we believe they may benefit from them:

"The conclusions of our analysis, we hope, will benefit the academic community and practitioners as well. For the former, the warnings we raised on the implications of the choice of the model to aggregate stakeholders' opinions may raise awareness among researchers working with similar methods, even in a different field; for the latter, the

| | conclusions of the analysis inform practitioners on the suitability of adoption one or more of the tools to achieve their goals of increasing awareness, social resilience and disaster-responsiveness." (Lines 513-517) |
|---|---|